# RegSegNet: A Joint Registration Segmentation Network for Automatic Liver Segmentation from Non-contrast 3D SPECT-CT Images

**Xuan Loc Pham**[1,2]                                    XUANLOC.PHAM@RADBOUDUMC.NL
[1] *Diagnostic Image Analysis Group, Radboud UMC, Nijmegen, The Netherlands*
[2] *VNU University of Engineering and Technology, 144 Xuan Thuy, Hanoi, Viet Nam*

**Manh Ha Luu**[2,3]                                        HA L4025@MED.CORNELL.EDU
[3] *Weill Cornell Medicine, Department of Radiology, New York, NY 10065, USA*

**Hong Son Mai**[4]                                       ALEX.HONG.SON@GMAIL.COM
[4] *Department of Nuclear Medicine, Hospital 108, Hanoi, Vietnam*

**Ngoc Ha Le**[4]                                          LENGOCHA108@YAHOO.COM
**Bram van Ginneken**[1]                             BRAM.VANGINNEKEN@RADBOUDUMC.NL

**Alessa Hering**[*1]                                   ALESSA.HERING@RADBOUDUMC.NL

## Abstract

3D SPECT-CT images play a vital role in the treatment process for liver cancer. However, in many cases, the CT scan taken alongside SPECT is non-contrast, making the liver segmentation task a tough challenge for both AI models and radiologists. Previous methods often faced trade-offs between accuracy and runtime. This study introduces RegSegNet, a deep learning model that utilizes image registration to effectively guide the liver segmentation in non-contrast SPECT-CT images. The proposed method is trained and evaluated on a dataset consisting of 60 liver cancer patients. Experimental results show that RegSegNet significantly outperforms baseline methods in terms of runtime while maintaining comparable accuracy.

**Keywords:** Joint Registration-Segmentation, Non-contrast SPECT-CT, Liver Cancer

## 1. Introduction

Liver cancer, or Hepatocellular carcinoma (HCC), is one of the leading causes of death worldwide and continues its worryingly increasing trend annually (Balogh et al., 2016). In the liver cancer treatment process, SPECT-CT is a popular medical imaging modality that effectively assists radiologists, particularly in the treatment planning phase. For instance, the liver volume segmentation from SPECT-CT images is crucial for optimizing dosimetry during Selective Internal Radiation Therapy (SIRT) treatment planning (Son et al., 2021). However, the utilization of contrast agents in the CT component of SPECT-CT images is often limited due to its potential adverse effects on patients' health. Without enhancement, distinguishing the liver boundary from adjacent organs becomes challenging, thereby posing a significant obstacle for both AI models and radiologists in segmenting the liver. (Luu et al., 2023)

Previously, several attempts have been made to segment the liver from non-contrast SPECT-CT images (nCECT). Tang et al. (2020) are among the first to apply deep learning to automatically segment the liver in a nCECT dataset with the proposal of a revised

---

* Corresponding author

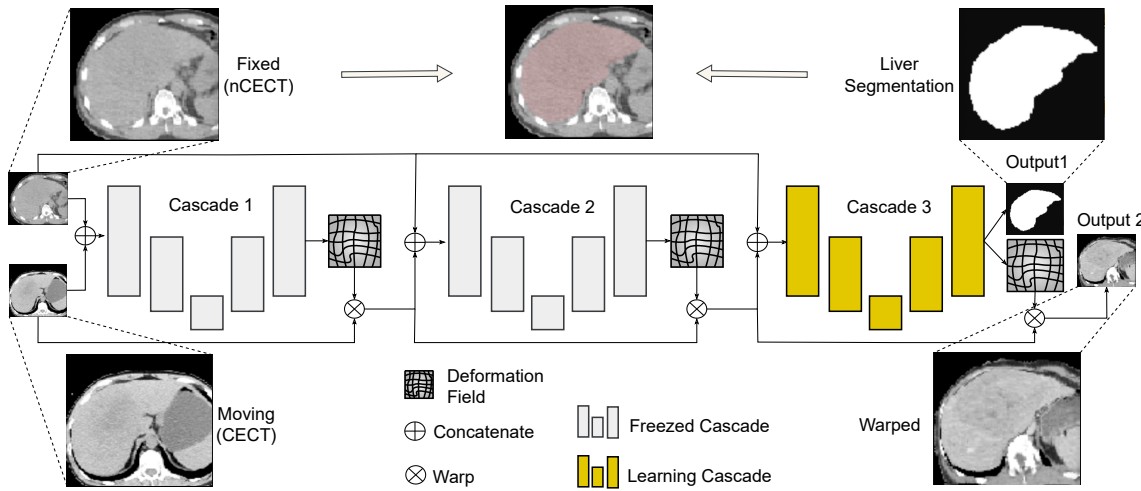

Figure 1: The model consists of three cascades, each utilizing a U-Net architecture. The first cascade generates a deformation field from a CECT-nCECT image pair, which is then used to update the moving image for subsequent cascades that refine registration progressively. Each cascade is independently trained per epoch, with prior cascades frozen and subsequent ones excluded.

Deepmedic architecture. Subsequently, Chaichana et al. (2021) introduced LiverNet, a CNN model that (semi)automatically delineates the liver in nCECT images. Both studies achieved a median Dice score of 91% in different datasets. More recently, Luu et al. (2023) advanced the field further with the proposal of a two-phase approach. The above study successfully leverages contrast-enhanced CT images (CECT) from the diagnostic phase in the liver cancer treatment process as an auxiliary reference for liver segmentation in nCECT images. However, the utilization of Elastix (Klein et al., 2009), a traditional image registration algorithm, made the method too time-consuming to be applied in clinical practice. In this study, we propose RegSegNet, a deep learning model that utilizes image registration to learn the alignments between pairs of CECT-nCECT, thus providing effective guidance for liver segmentation in nCECT images.

## 2. Methods

Recent studies have demonstrated the potential of deep learning-based approaches in liver CT image registration (Pham et al., 2024, 2023, 2022). Based on those ideas, we propose a novel multi-cascade architecture for RegSegNet as visualized in Figure 1. The model takes a pair of CECT-nCECT as input. Each cascade learns the misalignment of its input pair, producing the deformation field and feeding its knowledge to the succeeding cascade. For every epoch, the preceding cascade is trained first, then frozen to train the according cascade. The backpropagation is controlled by a combination of registration loss (Mutual

Table 1: Comparison of RegSegNet and baseline methods in various metrics

| Method | DSC | FNR | FPR | MSD (mm) | Run time |
|---|---|---|---|---|---|
| **Revised Deepmedic** | 0.91±0.03 | **0.03±0.01** | 0.14±0.05 | 143.16±61.7 | 27.8 sec |
| **Elastix + CNN** | **0.93±0.01** | 0.07±0.02 | **0.06±0.03** | **32±26** | ~122 sec |
| **RegSegNet** | 0.92±0.02 | 0.07±0.02 | 0.08±0.03 | 70.82±11.55 | **0.8 sec** |

Information loss), segmentation loss (Dice loss) and regularization loss (Bending Energy loss). The output is a set of liver segmentation masks and warped images.

## 3. Experiments and Results

The proposed method is trained and evaluated on the H108 dataset (Luu et al., 2023), comprising 60 liver cancer patients in the period between 2017 and 2021. 35 patients were allocated for training, 5 for validation and 20 for testing. The dataset includes diagnostic CECT, $^{99m}$Tc–MAA SPECT and nCECT image data. The liver annotations in the H108 dataset were initially performed manually by a technician and subsequently corrected or verified by two medical experts. For the training of RegSegNet, we mixed 35 patients into pairs of inter-patient and intra-patient as input, while the test dataset only contains pairs of intra-patient images. In addition, we also pre-trained baseline methods on the MSD dataset (Antonelli et al., 2022) before training on the H108 dataset.

Experimental results from Table 1 show that RegSegNet achieves a runtime of 0.8s, which is considerably faster than baseline methods. Regarding the Dice Similarity Coefficient (DSC), False Negative Rate (FNR) and False Positive Rate (FPR) metrics, there is only a slight decrease in performance. However, the two-phase approach outperforms RegSegNet in the Mean Surface Distance (MSD) metric.

## 4. Conclusions and Future Work

In summary, this study proposes leveraging CECT scans as supplementary reference information for liver segmentation in nCECT images. The registration component learns the correspondence between the contrast-enhanced liver and the non contrast-enhanced liver. The segmentation component relies on this correspondence to accurately segment the liver from the nCECT image with reference to the liver in the CECT image. This seamless integration of the registration and segmentation components into an end-to-end deep learning network helps considerably reduce the runtime compared to baseline methods while maintaining comparable accuracy in metrics such as DSC, FNR and FPR. However, this integration technique is outperformed by the two-phase approach in terms of the MSD metric. In future work, we will optimize the weight distribution among loss functions and also pre-train RegSegNet on the MSD dataset to further improve the performance of the proposed method.

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
