# OpenReview forum: "RegSegNet: A Joint Registration Segmentation Network for Automatic Liver Segmentation from Non-contrast 3D SPECT-CT Images"
_MIDL.io/2024/Short_Papers — MIDL 2024 Short Papers_

### Official Review · Reviewer_EgaZ · 2024-04-24

**Confidence:** 3
**Final Rating:** 3.5

**Review:**

Strength:
This paper introduced RegSegNet to utilize image registration to guide the liver segmentation in non-contrast SPECT-CT images. Experimental results demonstrated less runtime and comparable accuracy.

Weakness:
The clinical motivation of RegSegNet is not clearly illustrated. The accuracy was decreased significantly in terms of MSD compared to Elastix+CNN.

Summary:
Although the proposed method shows feasibility in practical application. The clinical motivation should be well justified.

---

### Decision · Program_Chairs · 2024-04-26

Accept